# A Knowledge-Based System as a Sustainable Software Application for the Supervision and Intelligent Control of an Alcoholic Fermentation Process

**Anca Sipos** 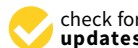

Faculty of Agricultural Sciences, Food Industry and Environmental Protection, Lucian Blaga University of Sibiu, 550012 Sibiu, Romania; anca.sipos@ulbsibiu.ro

**Abstract:** One goal of specialists in food processing is to increase production efficiency in accordance with sustainability by optimising the consumption of raw food materials, water, and energy. One way to achieve this purpose is to develop new methods for process monitoring and control. In the winemaking industry, there is a lack of procedures regarding the common work based on knowledge acquisition and intelligent control. In the present article, we developed and tested a knowledge-based system for the alcoholic fermentation process of white winemaking while considering the main phases: the latent phase, exponential growth phase, and decay phase. The automatic control of the white wine's alcoholic fermentation process was designed as a system on three levels. Level zero represents the measurement and adjustment loops of the bioreactor. At the first level of control, the three phases of the process are detected functions of the characteristics of the fermentation medium (the initial substrate concentration, the nitrogen assimilable content, and the initial concentration of biomass) and, thus, functions on the phase's duration. The second level achieves the sequence supervision of the process (the operation sequence of a fermentation batch) and transforms the process into a continuous one. This control level ensures the quality of the process as well as its diagnosis. This software application can be extended to the industrial scale and can be improved by using further artificial intelligence techniques.

**Keywords:** winemaking; alcoholic fermentation process; knowledge-based system; sustainable intelligent control

## 1. Introduction

Currently, the world is facing an unprecedented set of challenges and difficult problems, such as those related to energy and the environment, business and economics, food security, health, international stability, and the future of the technology, without knowing the long term effects on us and our planet [1,2].

Under these circumstances, food technology is a dynamically developing area in applied research and industry. One goal of specialists in food processing is to increase production efficiency in accordance with environmental preservation by optimising the consumption of raw food materials, water, and energy. There are different methods to achieve this purpose, such as improving and optimizing existing processes, using sustainable equipment and innovation in processes, and developing new methods for process monitoring and control [3,4].

In 2011, the International Organization of Vine and Wine defined the sustainable viticulture and winemaking industry as a "global strategy on the scale of the grape production and processing systems, incorporating at the same time the economic sustainability of structures and territories, producing quality products, considering requirements of precision in sustainable viticulture, risks to

the environment, products safety and consumer health and valuing of heritage, historical, cultural, ecological and landscape aspects" [5].

Among the different sectors of the food industry, the winemaking industry is of interest due to the continuous growth of the global market and production. Wine fermentation can be considered a transformation of must sugars into ethanol and $CO_2$; however, the process is more complex than this short definition. The small amounts of hundreds of different compounds that define most of the final organoleptic characteristics of the wine are based on biological, chemical, and physical processes that occur simultaneously [6,7].

The experimental and industrial processes drawn from winemaking observations have led to the conclusion that the first step in fermentation is catalyst synthesis, namely the exponential growth of the yeast with the nitrogen amount as limiting substrate, and the second step is catalysis, namely the degradation of the non-limiting sugar substrate into ethanol and carbon dioxide [8].

Due to the raw material variability and to the technological alcoholic fermentation process from white winemaking in general, researchers recommended that the automation control of this process be achieved via a knowledge-based system [9].

The development of modern food enterprises, and particularly winemaking enterprises, is inseparable from innovation [10]. An important strategic resource and an irreplaceable ingredient for enterprises to achieve innovation is a knowledge-based system [11]. A knowledge-based system is often an alternative to an expert system that refers to the expertise of human experts' knowledge. In the food safety industry, the term knowledge-based expert system is used with different meanings. The knowledge-based expert system can be assimilated with a computer programme capable of simulating human expert-like reasoning and decision-making within a special domain of expertise [12] or, a programme able to deal with processing information and mimic human experts [13] but, a broad sense of it is to be used as a technical solution supporting decision making [14]. The expertise of the knowledge-based system is obtainable with constant performance, as well as condensed, permanent, locally unbound, and immediate, in comparison with human experts. When planning an automation solution, a knowledge-based system saves time and money and improves the quality and scalability. A difficult task when creating a knowledge-based configurator is that the expert knowledge must be found, extracted, pre-processed, standardized, and condensed [1,15]. The knowledge-based system has been implemented with great success elsewhere in the manufacturing industry (i.e., the automotive industry, the energy industry, the chemical industry, in biotechnological processes) but not widely in the food industry which brings concerns to the food practitioners [16]. In the winemaking industry, there is a lack of procedures for collaborative work on knowledge acquisition and modelling, especially for multidisciplinary configuration projects, such as in food processing [17]. At this moment, the winemaking industry automatic control consists of measurement and adjustment loops of the important variables along the technological chain with the aim of ensuring an energy-saving and safe product [18]. In wine-producing countries, there should be increased the research collaboration efforts in both the industry and higher education, to reach a top-level of competency and awareness of their industrial potential. According to [19], the more developed countries are positioned at an important level of knowledge and use the available models and the knowledge-based systems. However, the small and medium-sized companies express a lack of knowledge and resources and consequently show limited use of these systems in the control process.

The present article investigated the design of a knowledge-based system for the alcoholic fermentation process of white winemaking, considering the main phases: the latent phase, exponential growth phase, and decay phase.

The automatic control objectives of this research were the following:

- finding the solution's phase determination of the fermentation, at any time of the process;
- the automation control of the process, depending on the length of each fermentation phase, to make decisions for improving the final quality of the wine (a more alcoholic wine, dryer, sweeter etc.) and to determine the end of the process;

- the diagnosis of the fermentation process.

The real-time information that led to the automation control was derived both from the process's physical transducers as well as from the state observers developed and presented in [20].

The current work novelty consists of developing a software application for the supervision, intelligent control, and data acquisition of the alcoholic fermentation process in white winemaking that can be implemented in industrial processing in the future.

## 2. Materials and Methods

The knowledge-based and the interface mechanisms were accomplished through the research undertaken for this paper (the automation control's reason for the fermentation process using the information from the knowledge-based mechanism was designed). For the development of the application, the supervision, control, and data acquisition software of the bioreactor from the research laboratory equipment were used. In Figure 1, the automation control structure is presented.

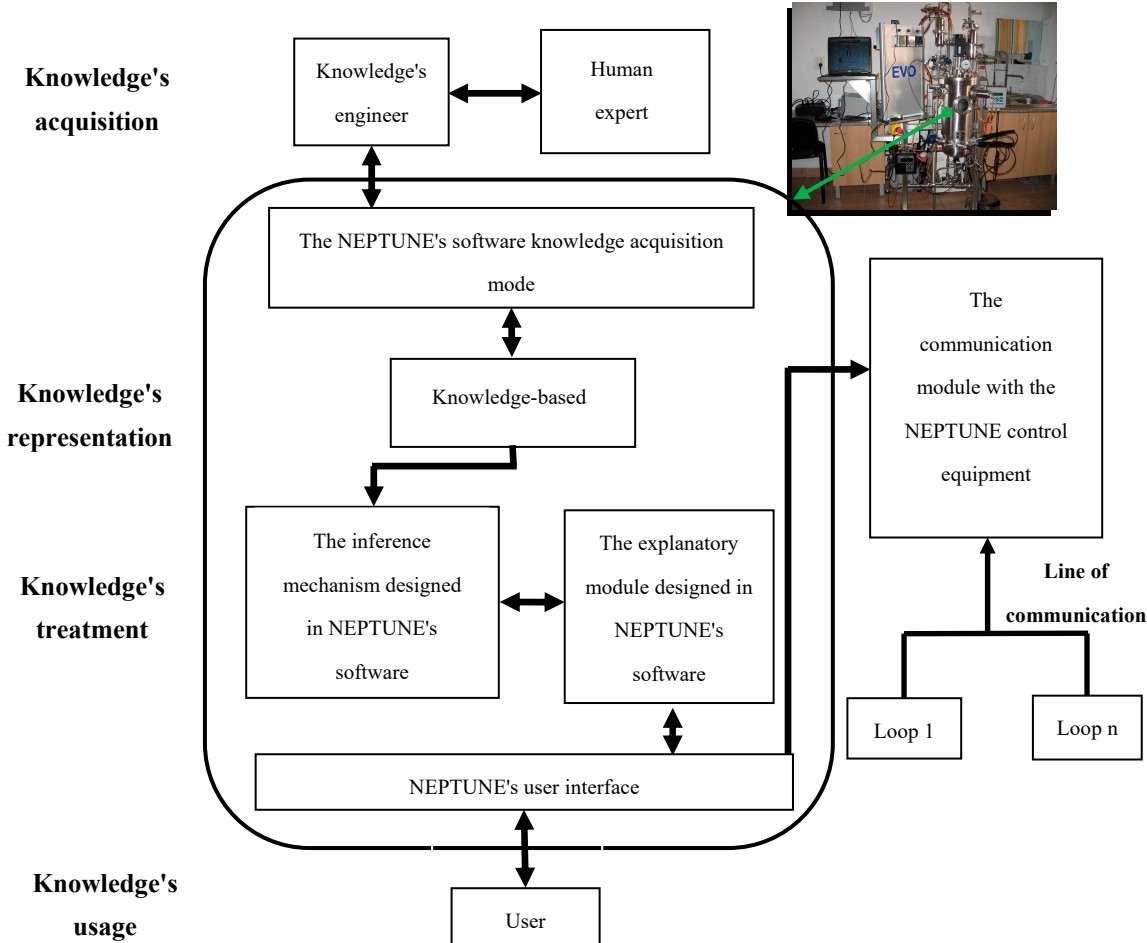

**Figure 1.** The scheme of the alcoholic fermentation process control system adapted from [21].

The main sources of knowledge applied in designing the knowledge-based control system were:

- the expert in the domain of the alcoholic fermentation processes;
- the information obtained from the case studies and from the experimental data;
- the specialised literature.

The expert in the field is the most important source of knowledge. An adequate method to obtain information from an expert is the interview approach. The interview is conducted in sessions,

involving an exchange of ideas on the issue such as the control conditions of the fermentation process, the physical, chemical, and microbiological variables of the initial conditions, the variables' correlations, which are of interest in the process control context etc.

The case studies are obtained pieces of information concerning the way in which the expert solves a real issue or by analyzing the experiments carried out previously. The experiments carried out in the research laboratory—modelling, simulation and advanced control of the technological processes and bioprocesses from food industry from "Lucian Blaga" University of Sibiu—were used for researching the alcoholic fermentation process of white winemaking. The literature data was used as a reference.

Specialised literature represents an additional source of knowledge. The existent documentation regarding the area of interest offers a global view of the issue and a clarification of the terminology specific to the domain. It has identified keywords and constructed search terms by examining recent literature reviews relating to the wine fermentation process (conditions, microbiological, chemical, and physical characteristic variables, models, process control) [4–6], and it has retrieved data from the Scopus and Web of Science libraries since the year 1989.

*2.1. Information on Alcoholic Fermentation Processes Obtained from a Representative Experimental Data Analysis*

As a result of the experience exchange with the experts in winemaking, ten datasets of "representative" experimental data were used for the analysis produced in the bioreactor in the research laboratory. The following findings resulted from the examination of the physical, chemical, and microbiological variables of the initial conditions, as well as of the control conditions:

1.  The investigation of the experimental data was conducted in four distinct control situations, presented in Table 1.
2.  Similar evolutions of the process were obtained under the same control conditions and, thus, of the monitored variables.
3.  Clues were identified in the experimental data regarding the process of the variables' correlations, which are of interest in the process control context. The following can be mentioned:

   - the temperature and the heat transfer variations affect the fermentation process quality (the alcohol concentration, flavourings, and by-product concentration) as well as the duration of fermentation [22];
   - the $CO_2$ quantity released during each phase of the fermentation process (especially in the tumultuous phase when the biomass is formed) provides relevant information regarding the evolution of the process's important variables (the substrate consumption, the biomass growth, the substrate deficiency in nitrogen and vitamins, and the alcohol production).

**Table 1.** The four types of fermentations realized at 21 °C.

| The Fermentation Type | Substrate | Yeast | The Adjusted Variables | The Measured and Adjusted Variables |
|---|---|---|---|---|
| a (mash malted) | mash malted | wine yeast: *Saccharomyces oviformis* and *Saccharomyces ellipsoideus* | temperature stirring speed | glucose concentration, yeast concentration, alcohol concentration, $CO_2$ released and dissolved concentration, $O_2$ released and dissolved concentration, pH |
| b (mash malted, B1) | mash malt enriched with B1 vitamin (thiamine) | wine yeast: *Saccharomyces oviformis* and *Saccharomyces ellipsoideus* | temperature stirring speed | glucose concentration, yeast concentration, alcohol concentration, $CO_2$ released and dissolved concentration, $O_2$ released and dissolved concentration, pH |

**Table 1.** *Cont.*

| The Fermentation Type | Substrate | Yeast | The Adjusted Variables | The Measured and Adjusted Variables |
|---|---|---|---|---|
| c (grape must) | white grape must | wine yeast: *Saccharomyces oviformis* and *Saccharomyces ellipsoideus* | temperature stirring speed | glucose concentration, yeast concentration, alcohol concentration, $CO_2$ released and dissolved concentration, $O_2$ released and dissolved concentration, pH |
| d (grape must, B1) | white grape must enrich with B1 vitamin (thiamine) | wine yeast: *Saccharomyces oviformis* and *Saccharomyces ellipsoideus* | temperature stirring speed | glucose concentration, yeast concentration, alcohol concentration, $CO_2$ released and dissolved concentration, $O_2$ released and dissolved concentration, pH |

### 2.2. The Automatic Control of the Alcoholic Fermentation Process

The automatic control of white wine's alcoholic fermentation process was designed as a three-level system. Level zero represents the measurement and adjustment loops of the bioreactor, regarding the pressure, the $CO_2$ and $O_2$ concentrations, the pH, the $pO_2$, the temperature, and the level. At the first level of control, the three phases of the process are the detected function of the characteristics of the fermentation medium (the initial substrate concentration, the nitrogen content that can be assimilated, the vitamin content, the initial concentration of biomass, and the type of yeasts) and, thus, functions of the phase's duration. The main variables determined at this level are the $CO_2$ concentration, the pH, and the optical density, which provides information regarding the total biomass concentration during the process. The second level achieves the sequence supervision of the process (the sequence of the operations of a fermentation batch) so that this level creates a continuous process. The second control level ensures the quality of the process, as well as its diagnosis.

### 3. Results and Discussion

Based on the experimental data obtained from the four types of fermentation, the following evolutions were observed: the pH, the optical density (the total concentration of the biomass), and the $CO_2$ realised concentration as shown in Figures 2–6.

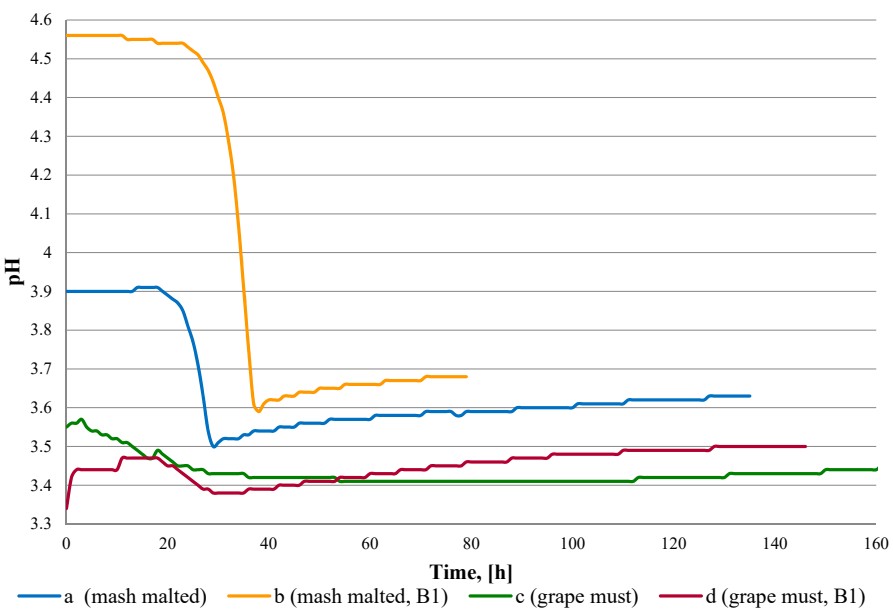

**Figure 2.** The evolution of the pH during the four types of fermentations.

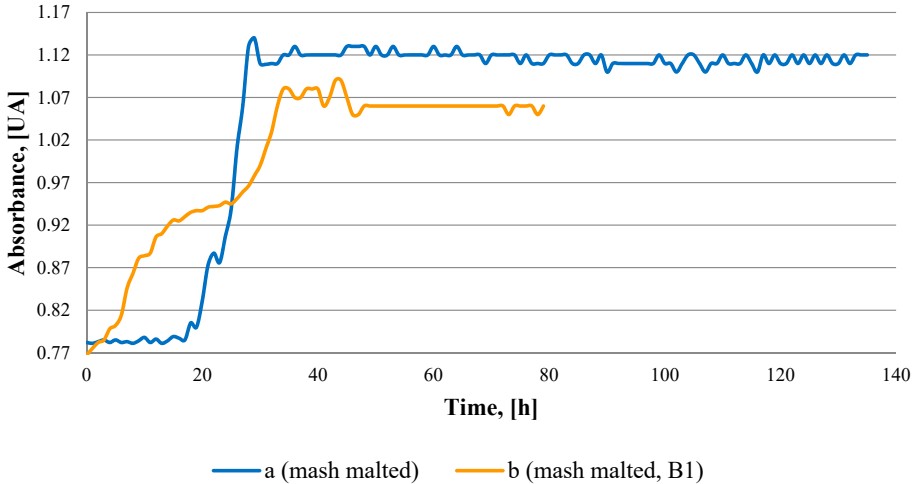

**Figure 3.** The optical density (the total concentration of the biomass) evolutions during the a and b types of fermentation.

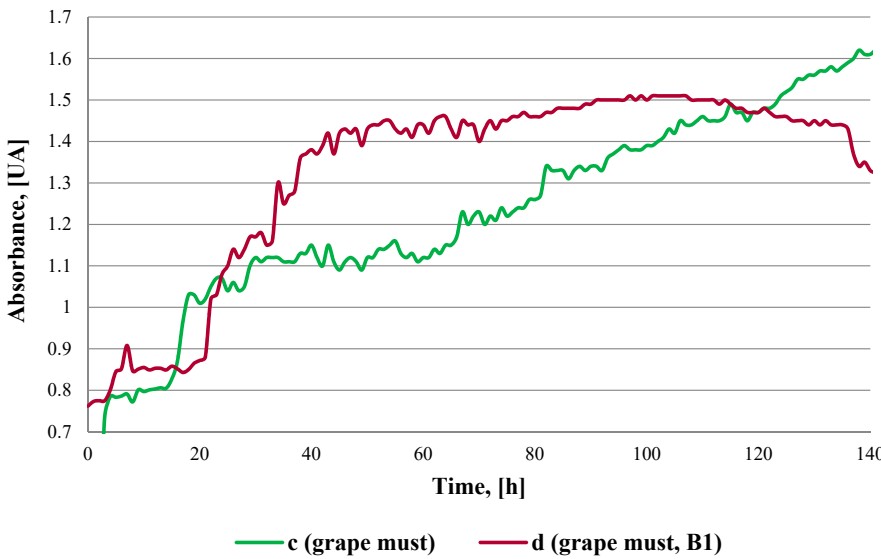

**Figure 4.** The optical density (the total concentration of biomass) evolution during types c and d of fermentation.

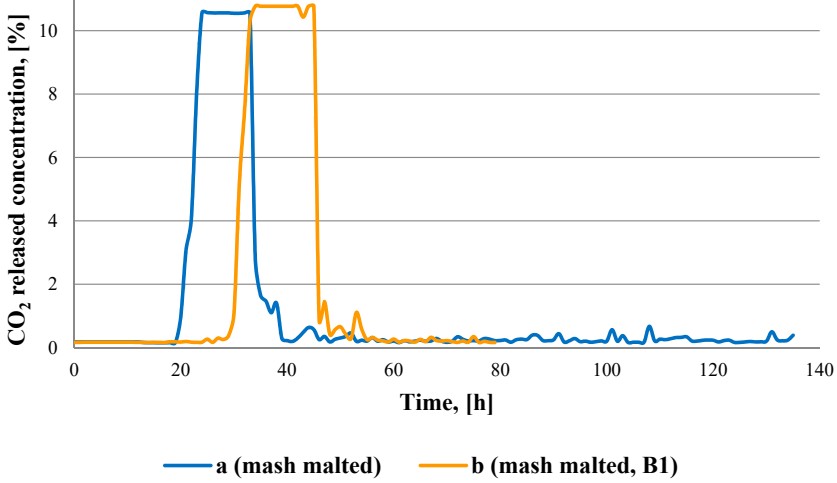

**Figure 5.** The $CO_2$ concentration evolution during the a and b types of fermentation.

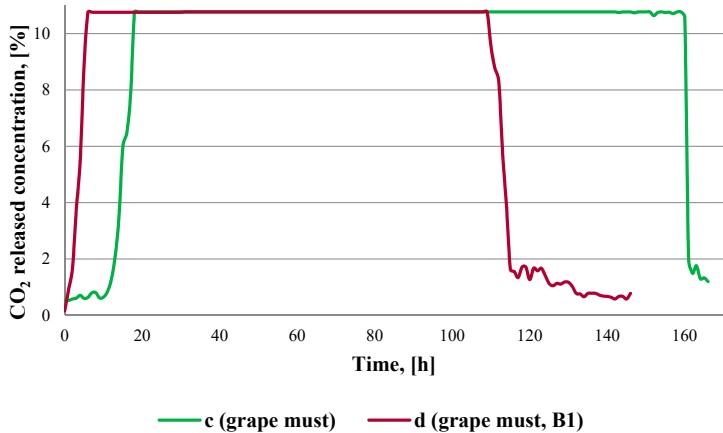

**Figure 6.** The $CO_2$ concentration evolutions during the c and d types of fermentations.

The results represented in Figures 2–6 led to the following conclusions:

- the variation of each studied variable (pH, optical density, and the $CO_2$ realised concentration), in all of the achieved fermentation cases, offers the possibility to determine the end and the start of the three phases of a fermentation process;
- the end and the beginning of the phases of the fermentation process are detectable based on rules that address the three variables monitored for each fermentation type and, therefore, the length of each phase; and
- by monitoring at least two of the three variables, the automatic control solution of the fermentation process can be developed.

### 3.1. The Automatic Control of the Alcoholic Fermentation at the First Level

The automatic control at this level must provide the phases detection of the fermentation process and, thus, the length of each phase. Therefore, using the knowledge obtained in the four types of fermentation, the first level is composed of rules that use data from the pH transducer, the $CO_2$ analyser, and the optical density sensor, as well as the extended estimator (for substrate concentration).

The structure of a rule is:

*RULE: rule_number*
*CF: certain factor*
*PRIO: priority*
*IF: premise*
*THEN: consequence*
*DESCRIPTION:*
where
*<premise>::=<logical _expression>*
*<logical_expression>::=(<variable_identifier> <relational_operator> <value>) AND/OR/NOT …*
*AND/OR/NOT (<variable_identifier> <relational_operator> <value>),*
and
*<consequence>::=(<variable_identifier>=<value>)AND/OR … AND/OR <variable_identifier>=<value>).*

Observations:

- the allowed relational operators are <, >, <=, >=, =, and <>;
- the priority rules for the logical operators AND, OR, and NOT are those from logical algebra.

Depending on the bioreactor equipment, we developed two types of rules for the three-phase identification of the fermentation process in accordance with Table 2.

**Table 2.** The automatic control rules at the first level.

| Variant | Rules |
|---|---|
| **I.**<br>- the bioreactor sensors used are the pH and $CO_2$ released concentrations<br>- extended Kalman filter for substrate concentration | **1_Rule:**<br>**IF** (($CO_2$ = const.) **AND** ($CO_2$ <= 0.5)) **AND** ((pH = const.) **OR** (pH decrease)) **AND** (substrate concentration = const.) **AND** (last_phase = latent_phase)<br>**THEN** (new_phase = latent_phase)<br>**DESCRIPTION:** identify the latent phase in biomass developing |
| | **2_Rule:**<br>**IF** (($CO_2$ rise) **OR** ($CO_2$ = const.)) **AND** (pH decrease) **AND** (substrate concentration decrease) **AND** ((last_phase = latent_phase) **OR** (last_phase = exponential_growth_phase))<br>**THEN** (new_phase = exponential_growth_phase)<br>**DESCRIPTION:** identify the exponential growth phase in biomass developing |
| | **3_Rule:**<br>**IF** (($CO_2$ decrease) **OR** (($CO_2$ = const.) **AND** ($CO_2$ <= 2))) **AND** (pH rise) **AND** (substrate concentration = const.) **AND** ((last_phase = exponential_growth_phase) **OR** (last_phase = decay_phase))<br>**THEN** (new_phase = decay_phase)<br>**DESCRIPTION:** identify the decay phase in biomass developing |
| **II.**<br>- the bioreactor sensors used are the concentrations of released pH and $CO_2$<br>- extended Kalman filter for substrate concentration<br>- extended Kalman filter for biomass concentration | **4_Rule:**<br>**IF** (($CO_2$ = const.) **AND** ($CO_2$ <= 0.5)) **AND** ((pH = const.) **OR** (pH decrease)) **AND** ((ABS = const.) **AND** (ABS <= 0.9 AU)) **AND** (substrate concentration = const.) **AND** (last_phase = latent_phase)<br>**THEN** (new_phase = latent_phase)<br>**DESCRIPTION:** identify the latent phase in biomass developing |
| | **5_Rule:**<br>**IF** (($CO_2$ rise) **OR** ($CO_2$ = const.)) **AND** (pH decrease) **AND** (((ABS rise) **OR** (ABS = const.)) **AND** (ABS > 1 AU)) **AND** (substrate concentration decrease) **AND** ((last_phase = latent_phase) **OR** (last_phase = exponential_growth_phase))<br>**THEN** (new_phase = exponential_growth_phase)<br>**DESCRIPTION:** identify the exponential growth phase in biomass developing |
| | **6_Rule:**<br>**IF** (($CO_2$ decrease) **OR** (($CO_2$ = const.) **AND** ($CO_2$ <= 2))) **AND** (pH rise) **AND** ((ABS = const.) **AND** (ABS > 1 AU)) **AND** (substrate concentration = const.) **AND** ((last_phase = exponential_growth_phase) **OR** (last_phase = decay_phase))<br>**THEN** (new_phase = decay_phase)<br>**DESCRIPTION:** identify the decay phase in biomass development |

*3.2. The Automatic Control of the Alcoholic Fermentation Process at the Second Level*

The automatic control of the fermentation process at this level has been accomplished in the form of data rules. The technologist controller is assisted by this data in the supervision and the control of the process.

The main functions of the automatic control at the second level are:

The control rules, which are divided into two subgroups:

a. The operation-sequencing of a fermentation process charge: this function contains all the operations necessary for a fermentation charge initiation: the bioreactor fed with the substrate, in site sterilization, inoculation with biomass, the fermentation process itself, and emptying and washing the bioreactor. Through correct sequencing of these operations and a process that is based on reality, the fermentation charge efficiency will be determined, thus, eliminating the deadlock that is inherent to a process controlled only by the technologist.

b. Rules regarding the achievement of the quality objectives, which are the fermentation stages employed in certain periods of time; these time spans depend on the type of yeast, on the fermentation temperature, and on the properties of the fermentation medium. The second level will intervene in each phase of the process, whenever the normal period of each phase is exceeded or reduced. In the latent phase, the following situation can appear: that it is too long either because the fermentation temperature is too low or due to fermentation medium deficiencies: the substrate concentration is too high for the type of yeast used or the biomass concentration is too high. In the exponential growth phase, two situations can occur: the fermentation is either

too turbulent (given either by too much heat or by the fermentation medium characteristics) or too slow (given either by a too low temperature or by fermentation medium deficiencies: the vitamin concentration is too small (especially the thiamine) or the assimilable nitrogen is too low).

The diagnosis rules are also divided into two categories:

a. The rules of state process monitoring, meaning the state process diagnosis. There are rules of cause detection that lead to overcoming the length of the fermentation process, alongside the monitoring of the bioreactor equipment's correct functioning (especially of the transducers) by providing the control function (the phase recognition based on a new rule), with a lower trust level (with an alert).

b. The second level intervention in the emergency situations: The early alert of an emergency or a potential situation arising can be envisaged (i.e., failing to achieve bioreactor sterilisation, non-feeding in time with the substrate and then with the biomass, not operating in accordance with the requirements of the temperature control loop). All these situations can lead to a compromise.

The fermentation process is defined following the sequencing rules of operations, as a finite automaton with a state number:

0 state—inactive bioreactor (but operational),

1 state—bioreactor in loading condition,

2 state—bioreactor in sterilisation condition,

3 state—bioreactor in cooling mode, and

4 state—bioreactor in fermentation process.

Moving from one state to another is done either by a control action of the technologist controller and/or under a signal action from a transducer.

In Table 3, the control rules of the two subgroups are presented.

**Table 3.** The control rules at the second automatic control level.

| Rules |
|---|
| **The control rules—the sequence of operations related to the fermentation process charge** |
| **7_Rule:**<br>**IF** (the process state is 0) **AND** (process' time = 0) **AND** (technologist controller' ordering = "loading")<br>**THEN** (the process state is 1)<br>**DESCRIPTION:** the bioreactor is programmed in loading condition |
| **8_Rule:**<br>**IF** (the process state is 1) **AND** (signal of complete loading = 1) **AND** (technologist controller'<br>ordering = "sterilisation")<br>**THEN** (the process state is 2)<br>**DESCRIPTION:** the bioreactor is ordered in sterilisation condition |
| **9_Rule:**<br>**IF** (the process state is 2) **AND** (signal of complete sterilisation = 1) **AND** (technologist controller' ordering = "the<br>fermentation mass' cooling")<br>**THEN** (the process state is 3)<br>**DESCRIPTION:** the bioreactor is ordered in fermentation mass' cooling mode |
| **10_Rule:**<br>**IF** (the process state is 3) **AND** (signal of complete cooling = 1) **AND** (technologist controller' ordering = "set the<br>references and start the control loops")<br>**THEN** (the process state is 0)<br>**DESCRIPTION:** the bioreactor is ordered in loading condition |

**Table 3.** *Cont.*

| Rules |
|---|
| **11_Rule:**<br>**IF** (the process state is 1) **AND** (signal of complete cooling = 1) **AND** (technologist controller's order = "execute the inoculation")<br>**THEN** (the process state is 4)<br>**DESCRIPTION:** the bioreactor is ordered in fermentation process |
| **12_Rule:**<br>**IF** (the process state is 4) **AND** (last_phase = decay_phase) **AND** (technologist controller's order = "determining the $t_{final\_optim}$")<br>**THEN** (the process state is 0)<br>**DESCRIPTION:** determining the fermentation process's shutdown time |
| **13_Rule:**<br>**IF** (the process state is 0) **AND** (signal of complete fermentation = 1) **AND** (technologist controller's order = "stop the bioreactor and evacuate the fermentation' mass")<br>**THEN** (the process state is 0)<br>**DESCRIPTION:** the bioreactor is ordered in evacuation mode |
| **14_Rule:**<br>**IF** (the process state is 0) **AND** (signal of complete evacuation = 1) B (technologist controller' ordering = "execute_washing_and_autoclaving_empty_ bioreactor")<br>**THEN** (the process state is 2)<br>**DESCRIPTION:** the bioreactor is ordered in sterilisation condition<br>The empty bioreactor sterilisation software has come up. |
| **15_Rule:**<br>**IF** (the process state is 2) **AND** (signal of complete sterilisation = 1) **AND** (the technologist controller orders = "execute shutdown bioreactor")<br>**THEN** (the process state is 0)<br>**DESCRIPTION:** the bioreactor is commanded into inactive state and a "good" charge is achieved |
| **The control' rules—the quality objectives achievement** |
| **16_Rule:**<br>**IF** (last_phase = latent_phase) **AND** ($\theta < \theta_{ref}$) **AND** ($t_{lat} > 20h$)<br>**THEN** (change the fermentation temperature setting point with up to maximum +3 °C)<br>**DESCRIPTION:** a discrepancy of the fermentation temperature with the used yeast appears |

| | |
|---|---|
| **I.**<br>- the bioreactor sensor used is pH<br>- extended Kalman filter for substrate concentration | **17_Rule:**<br>**IF** (last_phase = exponential_growth_phase) **AND** (pH does not drop) **AND** (substrate concentration does not drop)<br>**THEN** (add nitrogen (ammoniacal nitrogen and nitrogen from $\alpha$ amine) as well as vitamins (B1))<br>**DESCRIPTION:** bad evolution of the fermentation process is indicated by the pH level; a possible deficiency of the fermentation medium |
| **II.**<br>- the bioreactor sensors used are pH and $CO_2$ released concentration<br>- extended Kalman filter for substrate concentration | **18_Rule:**<br>**IF** (last_phase = exponential_growth_phase) **AND** (pH does not drop) **AND** (substrate concentration does not drop) **AND** ($CO_2$ does not grow)<br>**THEN** (add nitrogen and vitamins)<br>**DESCRIPTION:** the fermentation process negative evolution is indicated by the pH correlated with the $CO_2$ released; a possible deficiency of the fermentation medium |
| **The diagnosis rules—process monitoring** | |
| **I.**<br>- the bioreactor sensor used is pH<br>- extended Kalman filter for substrate concentration | **19_Rule:**<br>**IF** (last_phase = latent_phase) **AND** ((pH = const.) **OR** (pH does not drop)) **AND** ((substrate concentration = const.) **OR** (substrate concentration does not drop)) **AND** ($t_{lat} > 20h$)<br>**THEN** (check the substrate and biomass concentrations)<br>**DESCRIPTION:** the fermentation process negative evolution is indicated by the pH; a possible inhibition of the biomass |

**Table 3.** *Cont.*

| Rules | |
|---|---|
| **II.**<br>- the bioreactor sensors used are pH and $CO_2$ released concentration<br>- extended Kalman filter for substrate concentration | **20_Rule:**<br>**IF** (last_phase = latent_phase) **AND** ((pH = const.) **OR** (pH does not drop)) **AND** ((substrate concentration = const.) **OR** (substrate concentration does not drop)) **AND** ($CO_2$ does not grow) **AND** ($t_{lat}$ > 20h)<br>**THEN** (check the substrate and biomass concentrations)<br>**DESCRIPTION:** the fermentation process negative evolution is indicated by the pH correlated with the $CO_2$ released; a possible inhibition of the biomass |
| **The diagnosis rules—the emergency situations** | |
| **21_Rule:**<br>**IF** (last_phase = latent_phase) **AND** ($\theta < \theta_{ref}$) **AND** ($t_{lat}$ > 20h)<br>**THEN** (check the temperature loop)<br>**DESCRIPTION:** an emergency situation caused by the temperature loop is indicated | |
| **22_Rule:**<br>**IF** ($n < n_{lim}$)<br>**THEN** (check the speed loop)<br>**DESCRIPTION:** an emergency situation caused by the speed loop is indicated | |
| where *θ* is the reference, and *n* is the engine speed, (rpm). | |

### 3.3. Implementation of the Software Application on a Fermentation Process Control

A process control application was created, using the supervision, control, and data acquisition software of the bioreactor. This software application contains rules from the second level (the control rules: the operation sequencing of the fermentation process charge) and those from the first level, as shown in Table 4.

**Table 4.** The rules used in the fermentation process control software application.

| Rules |
|---|
| **The Control Rules—the Operation Sequencing of a Charge** |
| **IF** (the process state is 0) **AND** (process' time = 0) **AND** (technologist controller's order = "loading")<br>**THEN** (the process state is 1)<br>**DESCRIPTION:** the bioreactor is ordered in the loading condition |
| **IF** (the process state is 1) **AND** (signal of complete loading = 1) **AND** (technologist controller's order = "sterilization")<br>**THEN** (the process state is 2)<br>**DESCRIPTION:** the bioreactor is ordered in the sterilisation condition |
| **Loaded bioreactor software sterilization occurrence** |
| **IF** (the process state is 2) **AND** (signal of complete sterilisation = 1) **AND** (technologist controller's order = "the fermentation mass cooling")<br>**THEN** (the process state is 3)<br>**DESCRIPTION:** the bioreactor is ordered in the fermentation mass's cooling mode |
| **IF** (the process state is 3) **AND** (signal of complete cooling = 1) **AND** (technologist controller's order = "set the references and start the control loops")<br>**THEN** (the process state is 0)<br>**DESCRIPTION:** the bioreactor is ordered in the loading condition |
| **IF** (the process state is 1) **AND** (signal of complete cooling = 1) **AND** (technologist controller's order = "execute the inoculation")<br>**THEN** (the process state is 4)<br>**DESCRIPTION:** the bioreactor is ordered in the fermentation process |
| **The rules for identifying the fermentation process phases step in** |
| **IF** (($CO_2$ = const.) **AND** ($CO_2$ <= 0.5)) **AND** ((pH = const.) **OR** (pH decrease)) **AND** ((ABS = const.) **AND** (ABS <= 0.9 AU)) **AND** (substrate concentration = const.) **AND** (last_phase = latent_phase)<br>**THEN** (new_phase = latent_phase)<br>**DESCRIPTION:** identify the latent phase in the developing biomass |

**Table 4.** *Cont.*

| Rules |
| --- |
| **IF** ((CO$_2$ rise) **OR** (CO$_2$ = const.)) **AND** (pH decrease) **AND** (((ABS rise) **OR** (ABS = const.)) **AND** (ABS > 1 AU)) **AND** (substrate concentration decrease) **AND** ((last_phase = latent_phase) **OR** (last_phase = exponential_growth_phase)) **THEN** (new_phase = exponential_growth_phase) **DESCRIPTION:** identify the exponential growth phase in the developing biomass |
| **IF** ((CO$_2$ decrease) **OR** ((CO$_2$ = const.) **AND** (CO$_2$ <= 2))) **AND** (pH rise) **AND** ((ABS = const.) **AND** (ABS > 1 AU)) **AND** (substrate concentration = const.) **AND** ((last_phase = exponential_growth_phase) **OR** (last_phase = decay_phase)) **THEN** (new_phase = decay_phase) **DESCRIPTION:** identify the decay phase in the developing biomass |
| **Is returned to the operations' sequencing rules** |
| **IF** (the process state is 4) **AND** (last phase = decay phase) **AND** (technologist controller's order = "determining the t$_{final\_optim}$") **THEN** (the process state is 0) **DESCRIPTION:** determining the fermentation process shutdown time |
| **IF** (the process state is 0) **AND** (signal of complete fermentation = 1) **AND** (technologist controller's order = "stop the bioreactor and evacuate the fermentation mass") **THEN** (the process state is 0) **DESCRIPTION:** the bioreactor is ordered into evacuation mode |
| **IF** (the process state is 0) **AND** (signal of complete evacuation = 1) **AND** (technologist controller's order = "execute_washing_and_autoclaving_empty_ bioreactor") **THEN** (the process state is 2) **DESCRIPTION:** the bioreactor is ordered into sterilisation mode The empty bioreactor sterilisation software appears. |
| **IF** (the process state is 2) **AND** (signal of complete sterilisation = 1) **AND** (technologist controller's order = "execute_shutdown_bioreactor") **THEN** (the process state is 0) **DESCRIPTION:** the bioreactor is commanded into inactive condition and results in a "good" charge |

In Figure 7, the technologist controller interface layout is presented. This layout contains switches for triggering or stopping the automation control of different variables, as well as for on-line visualization, in a graphical form. The evolution of the monitored and controlled variables of the fermentation process can be observed. Here, the set-points are inserted, and the control loops of the bioreactor are activated.

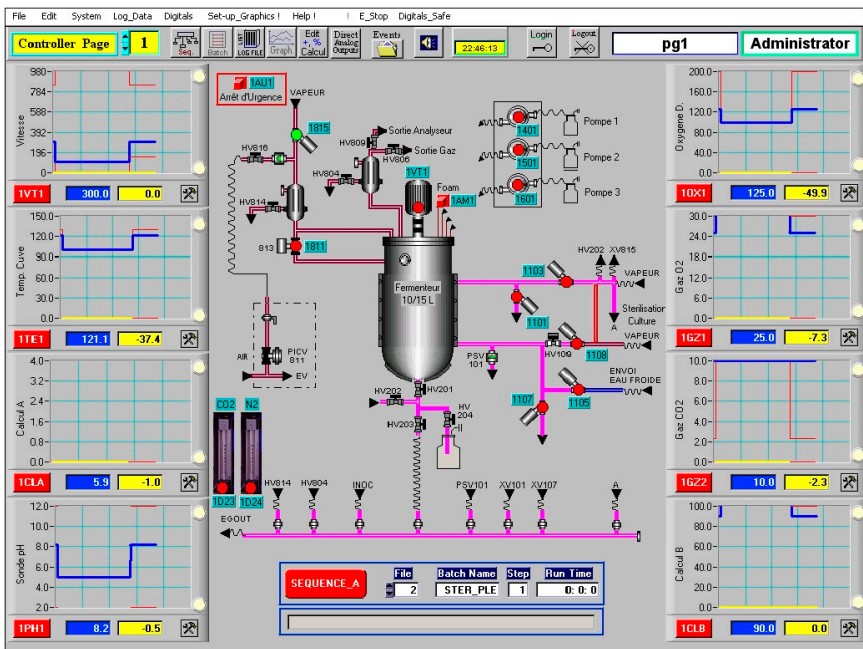

**Figure 7.** The interface layout of the technologist controller.

In the menu bar or tools bar, the wanted sequence of the control process is opened and saved as a file, SEQUENCE_C (Figure 8). Figure 9 contains the question list and the messages sent to the technologist controller. The software list of questions is represented by the control rules.

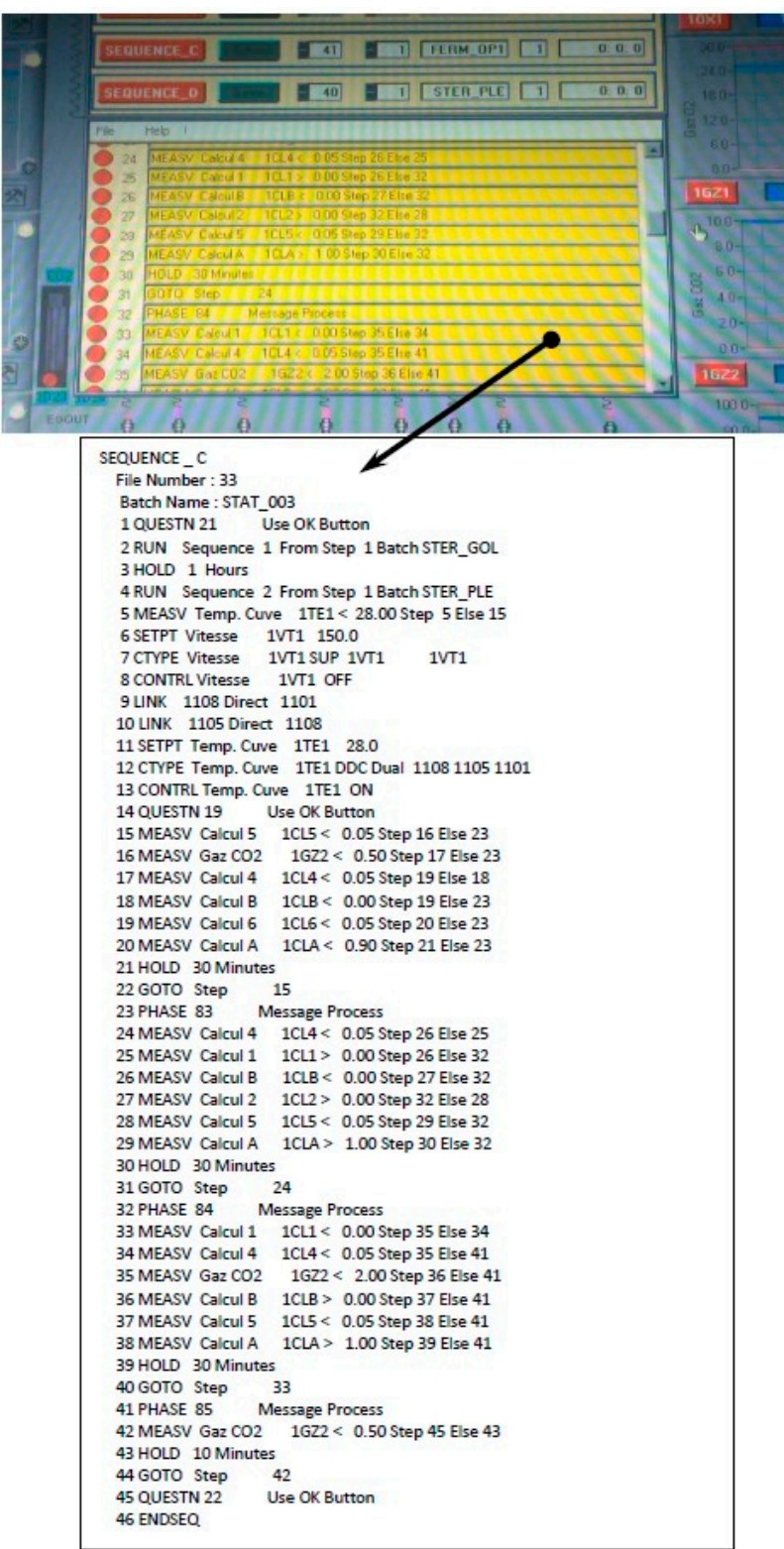

**Figure 8.** The set-off window of the sequence process control and the script window.

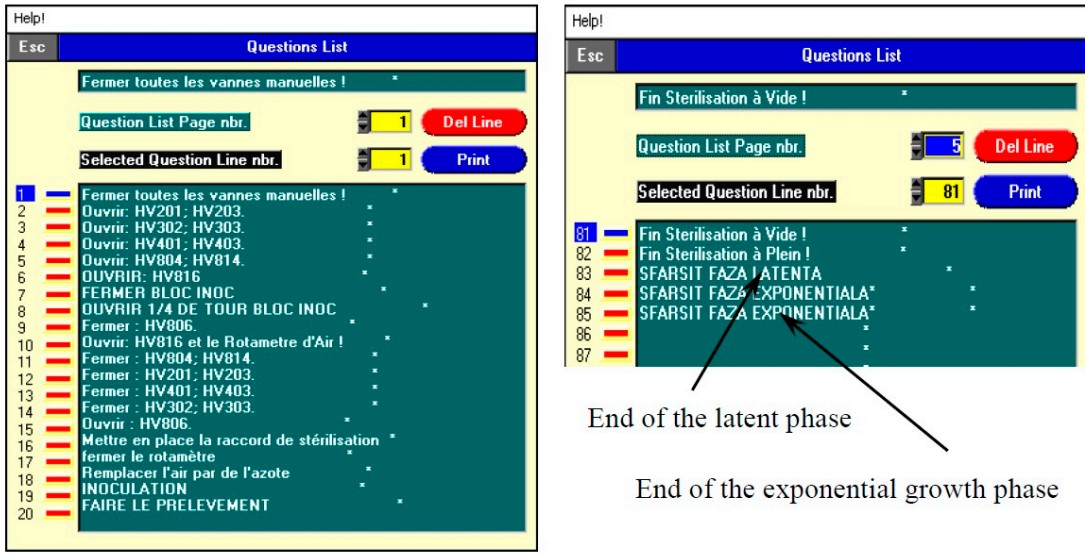

**Figure 9.** The windows which contain the question lists formed for the control rules and the messages sent to the technologist controller.

In Figure 10, the variables' trajectories show the evolution of the alcoholic fermentation process conducted at 20 °C, which was obtained under the software application.

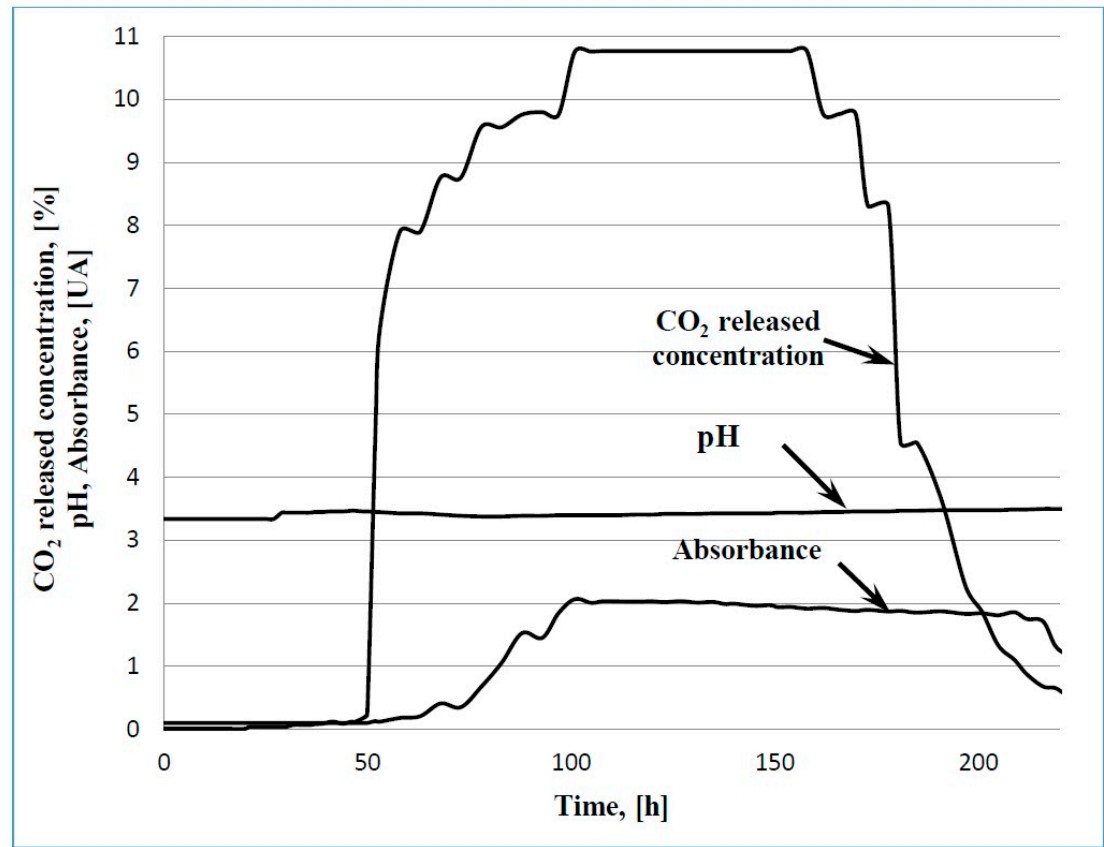

**Figure 10.** The variables' trajectories of the alcoholic fermentation process carried out at 20 °C.

By using and developing this process of control for several types of fermentation processes in winemaking and for processes that are entirely technological, both the safety and the quality of the

product can be provided. This can also assure a future for competitive and sustainable processing at all levels.

## 4. Conclusions

The need to develop a new and sustainable technological process in the winemaking industry will be guided by the cost-efficiency of processing and the price competitiveness of the final products. In accordance with this goal, the presented research led to an advanced control system of the alcoholic fermentation batch process in white winemaking through a knowledge-based system.

The main sources of knowledge applied in designing the knowledge-based control system were the experience exchange with the experts in winemaking regarding the physical, chemical, and microbiological variables of the initial conditions, the control conditions, and the information obtained from the case studies and from the experimental data. We identified clues in the experimental data regarding the process of the variables' correlations such as the temperature and the heat transfer variations, the $CO_2$ quantity released during each phase of the fermentation process and the pH variation. The automatic control of the process was designed as a system on three main levels. Level zero was represented by the measurement and adjustment loops of the bioreactor. At the first level of control, a system of rules was generated, which used information from pH transducers, $CO_2$ realised analysers, and optical density sensors, as well as from extended Kalman filters for the substrate concentration to detect the fermentation process phases and their duration. The second level, in the form of a finite automaton with a certain number of states, assists the technologist controller in supervision and control, as well as in assuring the qualitative variables of the alcoholic fermentation process. A process control application was created, using the supervision, control, and data acquisition software of the bioreactor. This software application contains rules from the second level (the control rules: the operation sequencing of the fermentation process charge) and those from the first level and was validated on an alcoholic fermentation process.

The implementation of the automation control as a bioreactor software application proved the possibility of the system being extended to the winemaking industrial scale (in the same time for large, small, and medium-sized companies) and can be adjusted for several areas of the food manufacturing sectors. In the future, this application can be improved by also using techniques provided by the synergy field, such as neural networks, genetic algorithms, intelligent control, and fuzzy logic. This automation and control system offers significant opportunities for the further integration of automated tools, methods, and technologies, which supports sustainable development.

**Funding:** This research received no external funding.

**Conflicts of Interest:** The authors declare no conflict of interest.

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
