# Peer review of "A Knowledge-Based System as a Sustainable Software Application for the Supervision and Intelligent Control of an Alcoholic Fermentation Process"

_sustainability, doi:10.3390/su122310205_

Round 1
Reviewer 1 Report
I am not an expert on wine's alcoholic fermentation processes, so I cannot assess the quality of the core of this proposal.
Nevertheless, I think this manuscript lacks a literature review. The author states:
In winemaking industries there is a lack of procedures for the collaborative work on knowledge acquisition and modeling, especially for multidisciplinary configuration projects, such as in food processing [12].
It could be true that there is a lack, but maybe the author could review other related areas. I doubt there are no other proposals similar to the proposal of this paper.
On the other hand, I also miss some references in some parts of the paper. For example, in:
In 2011, the International Organization of Vine and Wine has defined the sustainable viticulture and winemaking industry as a “global strategy on the scale of the grape production and processing systems, incorporating at the same time the economic sustainability of structures and territories, producing quality products, considering requirements of precision in sustainable viticulture, risks to the environment, products safety and consumer health and valuing of heritage, historical, cultural, ecological and landscape aspects”.
A reference to this statement of the International Organization of Vine and Wine is needed.
Or in:
The literature data was used as a reference.
Which literature data?
And, of course, in:
Specialised literature represents an additional source of knowledge. The existent documentation regarding the area of interest offers a global view of the issue and a clarification of the terminology specific to the domain.
Again, which literature?
Regarding the method, how the knowledge from the experts and from the specialised literature are combined?
An important flaw of this proposal is the lack of a validation and comparison with other alternative methods or systems. How is the author sure that her proposal is really a good one?
In Figure 1, I recommend the author to remove the background image, it makes difficult to read and understand the diagram.
In the charts, the different types of ferrmentations are labelled as a, b, c, and d. This labels make difficult to understand the charts. I recommend the authors to use more significant labels such as "a (mash malt), b (mash malt B1), ..."
The reference list should be checked. For example, I have found some issues with the DOI, different formats, for example:
10.3182/20110828-6-IT-1002.01189
https://doi.org/10.3390/su12177105
doi:10.3390/foods9010033
The English language is fine, I think I have just found one error in the phrase:
31 without knowing their long term affects on us and our planet [1,2].
I think the author wants to says "their long term effects".
Author Response
Nevertheless, I think this manuscript lacks a literature review. The author states:
In winemaking industries there is a lack of procedures for the collaborative work on knowledge acquisition and modeling, especially for multidisciplinary configuration projects, such as in food processing [12].
It could be true that there is a lack, but maybe the author could review other related areas. I doubt there are no other proposals similar to the proposal of this paper.
Response of the author:
The demanded requests were performed in the manuscript. The added text is in red.
On the other hand, I also miss some references in some parts of the paper. For example, in:
In 2011, the International Organization of Vine and Wine has defined the sustainable viticulture and winemaking industry as a “global strategy on the scale of the grape production and processing systems, incorporating at the same time the economic sustainability of structures and territories, producing quality products, considering requirements of precision in sustainable viticulture, risks to the environment, products safety and consumer health and valuing of heritage, historical, cultural, ecological and landscape aspects”.
A reference to this statement of the International Organization of Vine and Wine is needed.
Response of the author:
The corrections requested was performed in the manuscript.
Or in:
The literature data was used as a reference.
Which literature data?
And, of course, in:
Specialised literature represents an additional source of knowledge. The existent documentation regarding the area of interest offers a global view of the issue and a clarification of the terminology specific to the domain.
Again, which literature?
Response of the author:
The demanded requests were performed in the manuscript. The added texts are in red.
Regarding the method, how the knowledge from the experts and from the specialised literature are combined?
Response of the author:
The demanded requests were performed in the manuscript. The added text is in red.
With the expert in the domain were an exchange of ideas on the issue such as: the control conditions of the fermentation process, the physical, chemical, and microbiological variables of the initial conditions, the variables’ correlations, which are of interest in the process control context etc. From the specialised literature were identified aspects regarding wine fermentation process: conditions, microbiological, chemical, and physical characteristic variables, models, process control proposed in the articles.
An important flaw of this proposal is the lack of a validation and comparison with other alternative methods or systems. How is the author sure that her proposal is really a good one?
Response of the author:
After the retrieved data from the Scopus and Web of Science libraries since the year 1989 we didn't find an article with the application of knowledge-based system in winemaking alcoholic fermentation process. We find some applications in bioprocesses from environmental protection but these fermentation processes have different aspects. As we affirm in the article: at this moment, the winemaking industry automatic control consists in measurement and adjustment loops of the important variables along the technological chain with the aim of ensuring a safety product and an energy saving.
In Figure 1, I recommend the author to remove the background image, it makes difficult to read and understand the diagram.
Response of the author:
The demanded request was performed in the manuscript.
In the charts, the different types of ferrmentations are labelled as a, b, c, and d. This labels make difficult to understand the charts. I recommend the authors to use more significant labels such as "a (mash malt), b (mash malt B1), ..."
Response of the author:
The demanded request was performed in the manuscript.
The reference list should be checked. For example, I have found some issues with the DOI, different formats, for example:
10.3182/20110828-6-IT-1002.01189
https://doi.org/10.3390/su12177105
doi:10.3390/foods9010033
Response of the author:
The demanded requests were performed in the manuscript.
The English language is fine, I think I have just found one error in the phrase:
31 without knowing their long term affects on us and our planet [1,2].
I think the author wants to says "their long term effects"
Response of the author:
The article was corrected the English language by a native specialist. I made the demanded requests in the corrected variant.

Reviewer 2 Report
The work presents interest for the food industry and for wine making industry. The article consists in the presentation of the design of an automatic control of the white wine’s alcoholic fermentation process which can optimize the process and can improve its profitability.
The introduction section could be improved by a deep research of the specialty literature, citation of other studies developed on this subject and comparation of the methods/software utilized.
The conclusions section can be improved by emphasizing the results of the study and the benefits that result from the utilization of this automatic control system in the wine making industry.
The English could be improved and the text from table 2 should be carefully revised, it still contain words in Romanian language.
Author Response
The introduction section could be improved by a deep research of the specialty literature, citation of other studies developed on this subject and comparation of the methods/software utilized.
The conclusions section can be improved by emphasizing the results of the study and the benefits that result from the utilization of this automatic control system in the wine making industry.
The English could be improved and the text from table 2 should be carefully revised, it still contain words in Romanian language.
Response of the author:
The demanded requests were performed in the manuscript. The added texts are in red. The English language' article was corrected by a native specialist. I made the demanded requests in the corrected variant.

Round 2
Reviewer 1 Report
I have reviewed the new version of the manuscript and I can confirm the authors have addressed all my previous comments. Therefore, I think the manuscript has improved a lot and is ready to be published.
I have only concerns about the figures, because they are not included in the manuscript. They can be found in an additional Word file, I think this is not going to be an issue.
Regarding the figures, I repeat my previous comment:
In the charts, the different types of ferrmentations are labelled as a, b, c, and d. This labels make difficult to understand the charts. I recommend the authors to use more significant labels such as "a (mash malt), b (mash malt B1), ..."
The authors have not improved the legends of the figures 2-6, so it is difficult to understand them. I recommend the authors to change those figures as Table 1.
Author Response
Response of the author:
The corrections requested were performed in the figures document.
